# CVE2ATT&CK: BERT-Based Mapping of CVEs to MITRE ATT&CK Techniques

Octavian Grigorescu [1], Andreea Nica [1], Mihai Dascalu [1,2,*] and Razvan Rughinis [1,2]

1 Computer Science & Engineering Department, University Politehnica of Bucharest, 313 Splaiul Independentei, 060042 Bucharest, Romania
2 Academy of Romanian Scientists, Str. Ilfov, Nr. 3, 050044 Bucharest, Romania
* Correspondence: mihai.dascalu@upb.ro

**Abstract:** Since cyber-attacks are ever-increasing in number, intensity, and variety, a strong need for a global, standardized cyber-security knowledge database has emerged as a means to prevent and fight cybercrime. Attempts already exist in this regard. The Common Vulnerabilities and Exposures (CVE) list documents numerous reported software and hardware vulnerabilities, thus building a community-based dictionary of existing threats. The MITRE ATT&CK Framework describes adversary behavior and offers mitigation strategies for each reported attack pattern. While extremely powerful on their own, the tremendous extra benefit gained when linking these tools cannot be overlooked. This paper introduces a dataset of 1813 CVEs annotated with all corresponding MITRE ATT&CK techniques and proposes models to automatically link a CVE to one or more techniques based on the text description from the CVE metadata. We establish a strong baseline that considers classical machine learning models and state-of-the-art pre-trained BERT-based language models while counteracting the highly imbalanced training set with data augmentation strategies based on the TextAttack framework. We obtain promising results, as the best model achieved an F1-score of 47.84%. In addition, we perform a qualitative analysis that uses Lime explanations to point out limitations and potential inconsistencies in CVE descriptions. Our model plays a critical role in finding kill chain scenarios inside complex infrastructures and enables the prioritization of CVE patching by the threat level. We publicly release our code together with the dataset of annotated CVEs.

**Keywords:** MITRE ATT&CK Matrix; techniques classification; BERT-based multi-labeling

## 1. Introduction

Cyberspace has become a fundamental component of everyday activities, being the core of most economic, commercial, cultural, social, and governmental interactions [1]. As a result, the ever-growing threat of cyber-attacks not only implies a financial loss, but also jeopardizes the performance and survival of companies, organizations, and governmental entities [2]. It is vital to recognize the increasing pace of cybercrime as the estimated monetary cost of cybercrime skyrocketed from approximately $600 billion in 2018 to over $1 trillion in 2020 [3]. This effect has increased even further due to the COVID-19 pandemic [4].

In this context, the necessity for better cyber information sources and a standardized cybersecurity knowledge database is of paramount importance, as a means to identify and combat the emerging cyber-threats [5]. Efforts to build such globally accessible knowledge bases already exist. MITRE Corporation set up two powerful public sources of cyber threat and vulnerability information, namely the Common Vulnerabilities and Exposures list and the MITRE ATT&CK Enterprise Matrix.

The *Common Vulnerabilities and Exposures* list is a community-based dictionary of standardized names for publicly known cybersecurity vulnerabilities. Its effort converges toward making the process of identifying, finding, and fixing software vulnerabilities more efficient, by providing a unified naming system [6]. Despite their benefits and widespread

usage, CVE entries offer little to no information regarding mitigation techniques or existing defense strategies that could be employed to address a specific vulnerability. Moreover, the meta-information of a CVE does not include sufficient classification qualities, resulting in sub-optimal usage of this database. Better classification would translate to mitigating a larger set of vulnerabilities since they can be grouped and addressed together [7].

The *MITRE ATT&CK Enterprise Matrix* links techniques to tangible configurations, tools, and processes that can be used to prevent a technique from having a malicious outcome [8]. By associating an ATT&CK technique to a given CVE, more context and valuable information for the CVE can be extracted, since CVEs and MITRE ATT&CK techniques have complementary value. Furthermore, security analysts could discover and deploy the necessary measures and controls to monitor and avert the intrusions pointed out by the CVE and cluster the CVEs by technique [9].

Even though linking CVEs to the MITRE ATT&CK Enterprise Matrix would add massive value to the cybersecurity community, these two powerful tools are currently separated. However, manually mapping all 189,171 [10] CVEs currently recorded to one or more of the 192 different techniques in the MITRE ATT&CK Enterprise Matrix is a non-trivial task and the need for automated models emerges to map all existing entries to corresponding techniques. In addition, even if new CVEs would be manually labeled, an initial pre-labeling using a machine learning model before expert validation would be time effective and beneficial. Moreover, the model would provide technique labeling for zero-day vulnerabilities, which would be extremely helpful for security teams.

The ATT&CK matrix supports a better understanding of vulnerabilities and what an attacker could achieve by exploiting a certain vulnerability. ATT&CK technique details, such as detection and mitigation, are useful for system administrators, SecOps, or DevSec-Ops teams to obtain an assessment risk report in a short period of time while generating a remediation plan for discovered vulnerabilities. The Center for Threat-Informed Defense team has created a very useful methodology [11] that helps the community build a more powerful threat intelligence database. The organization's defender team has to understand how important it is to bridge vulnerability and threat management with the adoption of this methodology as more reliable and consistent risk assessment reports will be obtained [12].

Baker [12] highlights the importance of combining CVEs with the ATT&CK framework to achieve threat intelligence. Years ago, it was considerably harder for security teams to understand the attack surface, thus reducing their capacity to protect the organization against cyber attacks. With the emergence of the ATT&CK project, the security teams have a better overview of the CVEs based on known attack techniques, tactics, and procedures.

Vulnerability management can be divided into three categories, namely: the "Find and fix" game, the "Vulnerability risk" game, and the "Threat vector" game. The first one is a traditional approach where the vulnerabilities are prioritized by CVSS Score; this is applicable for small organizations with less dynamic assets. The second category consists of risk-based vulnerability management where organizational context and threat intelligence (such as CVE exploited in the wild properties) are considered; this applies to organizations that have security teams, but the number of CVEs is too large. The "Threat Vector" game includes the understanding of how the hackers might exploit the vulnerabilities while accounting for the MITRE ATT&CK framework mappings between CVEs and techniques, tactics, and procedures. The third category is the most efficient model of threat intelligence, with inputs delivered to the vulnerability risk management process from cyber attacks that have occurred and are trending. As such, security teams should take into account risks for building the vulnerability management program, but also threat intelligence to have a better understanding of vulnerabilities and to discover the attack chains within the network [13].

The aim of this paper is to develop a model that leverages the textual description found in CVE metadata to create strong correlations with the MITRE ATT&CK Enterprise Matrix techniques. To achieve this goal, a data collection methodology is developed to build our manually labeled CVE corpus containing more than 18,100 entries. Moreover, state-of-the-

art Natural Language Processing (NLP) techniques that consider BERT-based architectures are employed to create robust models. We also target addressing the problem of a severely imbalanced dataset by developing an oversampling method based on adversarial attacks.

Efforts have been already undertaken to interconnect CVEs to the MITRE ATT&CK Framework. However, we identified limitations of existing solutions based on the research gap in the literature regarding the identification of correspondences between CVEs to the corresponding techniques from the MITRE ATT&CK Enterprise Matrix. The following subsections details existing state-of-the-art techniques relevant for our task.

### 1.1. BRON

BRON [9] is a bi-directional aggregated data graph which allows relational path tracing between MITRE ATT&CK Enterprise Matrix tactics and techniques, Common Weakness Enumerations (CWE), Common Vulnerabilities and Exposures (CVE), and Common Attack Pattern Enumeration and Classification list (CAPEC). BRON creates a graph framework that unifies all scattered data through inquiries performed of the resulted graph representation by data-mining the relational links between all these cyber-security knowledge sources. In this manner, it connects the CVE list to MITRE ATT&CK by traversing the relational links in the resulted graph.

Each information source has a specific node type, interconnected by external linkages as edges. MITRE ATT&CK techniques are linked to Attack Patterns. Attack Patterns are connected to CWE Weaknesses, which have relational links to a CVE entry. Thus, BRON can respond to several different queries, including linking the CVE list to the MITRE ATT&CK Framework.

However, the model falls short as it does not connect new CVEs to MITRE ATT&CK Enterprise Matrix techniques, but it uses already existing information and links to create a more holistic overview of the already available knowledge. It does not solve our problem, since the main aim is to correctly label new emergent samples.

### 1.2. CVE Transformer (CVET)

The CVE Transformer (CVET) [14] is a model that combines the benefits of using the pre-trained language model RoBERTa with a self-knowledge distillation design used for fine-tuning. Its main aim is to correctly associate a CVE with one of 10 tactics from the MITRE ATT&CK Enterprise Matrix. Although the CVET approach obtains increased performance in F1-score, it is unable to identify all 14 tactics from the MITRE ATT&CK Matrix on the training knowledge base.

Moreover, the problem of technique labeling is much more complex than tactic mapping, since the number of available techniques is ten times higher (i.e., there are 14 tactics and 192 different techniques in the MITRE ATT&CK Enterprise Matrix). Additionally, tactic labeling can be viewed as a subproblem of our main goal given the correlation between tactics and techniques. Overall, technique labeling is out of scope for the CVE Transformer project.

### 1.3. Unsupervised Labeling Technique of CVEs

The unsupervised labeling technique introduced by Kuppa et al. [15] considers a multi-head deep embedding neural network model that learns the association between CVEs and MITRE ATT&CK techniques. The proposed representation identifies specific regular expressions from the existing threat reports and then uses the cosine distance to measure the similarity between ATT&CK technique vectors and the text description provided in the CVE metadata. This technique manages to map only 17 techniques out of the existing 192. As such, multiple techniques are not covered by the proposed model. Thus, a supervised approach for technique labeling might improve the recognition rate among techniques.

*1.4. Automated Mapping to ATT&CK: The Threat Report ATT&CK Mapper (TRAM) Tool*

Threat Report ATT&CK Mapping (TRAM) [16] is an open-source tool developed by *The Center for Threat-Informed Defense* that automates the process of mapping MITRE ATT&CK techniques on cyber-threat reports. TRAM utilizes classical pre-processing techniques (i.e., tokenization, stop-words removal, lemmatization) [17] and applies Logistic Regression on the bag-of-words representations. Since the tool maps any textual input on MITRE ATT&CK techniques, it could, in theory, be adapted to link the CVE list to the MITRE ATT&CK Framework by simply using it on the CVE textual description. However, due to its simplicity, the tool has serious limitations when it comes to its capacity to learn the right association between text descriptions and techniques. In addition, TRAM labels each sentence individually, failing to capture dependencies in textual passages. In this way, the overall meaning of the text is lost.

The main contributions of this paper are as follows:

- Introducing a new publicly available dataset of 1813 CVEs annotated with all corresponding MITRE ATT&CK techniques;
- Experiments with classical machine learning and Transformer-based models, coupled with data augmentation techniques, to establish a strong baseline for the multi-label classification task;
- A qualitative analysis of the best performing model, coupled with error analysis that considers Lime explanations [18] to point out limitations and future research directions.

We open-source our dataset on TagTog [19] and the code on GitHub [20].

## 2. Method

This section provides an overview of our proposed methodology, focusing on: (1) data collection and building the corpus needed for training the models; and (2) exploring various neural architecture for mapping CVEs to ATT&CK techniques.

*2.1. Our Labeled CVE Corpus*

2.1.1. Data Collection

Since no public datasets exist that map a CVE to all corresponding ATT&CK techniques, the first step consisted of building our own labeled corpus of 1813 CVEs, which was obtained using two different methods.

First, we manually created a knowledge base of 993 labeled CVEs by individually mapping each CVE to tactics and techniques from MITRE ATT&CK Enterprise Matrix. We extracted CVEs that were published between 2020 to 2022 for relevance. The labeling process was performed by 4 experts to ensure consistency, following the standardized approach proposed by the *Mapping MITRE ATT&CK to CVEs for Impact* methodology [11] and a set of common general guidelines.

The *Mapping MITRE ATT&CK to CVEs for Impact* methodology consists of three steps. The first one is to identify the type of vulnerability (e.g., cross-site scripting, buffer overflow, SQL injection) based on the vulnerability type mappings. The next step is to find the functionality to which the attacker gains access by exploiting the CVE. The final step refers to determining the exploitation technique using the provided tips that offer details about the necessary steps to exploit a vulnerability. Our methodology started from these steps and added other common general guidelines before labeling the tactics and techniques, such as searching for more details about a CVE on security blogs to obtain more relevant insights, or analyzing databases (e.g., the Vulnerability Database [21] and the Exploit Database—Exploits for Penetration Testers, Researchers, and Ethical Hackers [22]) for useful inputs about CVEs.

The labeling was performed by three 4th year undergraduate students in Computer Science with background courses in security, networking, and operating systems, and one Ph.D. student in Computer Science with 5+ years of experience in information security in the industry who provided guidance and helped reach consensus. The entire annotation

process was overseen by a professor in cyber security. The dataset can be found on TagTog [19] and is split into the following collections:

1.  *Inter-rater*—A collection of 24 CVEs evaluated by all experts to ensure high agreement and consistent annotations; this collection was used for training the raters until perfect consensus was achieved;
2.  *Double-rater*—A collection of 295 CVE evaluated by pairs of two raters; this collection was created after some experience was accumulated and consensus among raters was achieved using direct discussions;
3.  *Individual*—A collection of 674 CVE evaluated by only one rater; this collection was annotated after the initial training phase was complete and raters gained experience.

Second, besides the manual labeling process, we automatically extracted 820 already labeled CVEs provided by *Mapping MITRE ATT&CK to CVEs for Impact* [11] and imported them in our TagTog project. The provided CVEs date from 2014 to 2019; thus, there is no overlap with the manually annotated CVEs.

Each CVE entry has associated the corresponding ID, the rich text description, and 14 labels denoting the possible tactics found in the MITRE ATT&CK Enterprise Matrix where the corresponding techniques are annotated. Extracting the data from TagTog can be performed automatically, using the TagTog API [23].

### 2.1.2. Data Analysis

The size of our corpus can be argued by the increased difficulty when annotating a CVE and the impossibility to find other previously build repositories consisting of CVEs mapped on MITRE ATT&CK Enterprise Matrix both tactics and techniques. As discussed previously, more than 189,171 CVEs currently exist and our dataset only captures a fraction of them. Moreover, the distribution of CVEs based on technique is highly imbalanced (see Figure 1) because the CVEs were collected based on their release date, without any other further considerations. About 77% of the collected CVEs cover 5 techniques (*Exploit Public-Facing Application*, *Exploitation for Client Execution*, *Command and Scripting Interpreter*, *Endpoint Denial of Service* and *Exploitation for Privilege Escalation*).

Figure 1 also shows that a large number of techniques contain a far too small number of examples for effective learning. As such, a threshold of a minimum of 15 examples per technique was imposed. In this manner, out of the 192 different techniques from the MITRE ATT&CK Enterprise Matrix, only 31 were considered in follow-up experiments. The CVEs that are not mapped to any of the 31 considered techniques were also discarded, leaving a total of 1665 annotated examples in the dataset. Figure 2 depicts the new distribution of CVEs based on technique after applying the threshold.

### 2.1.3. Data Augmentation

The severe data imbalance which characterizes our CVE dataset can potentially degrade the performance of many machine learning models since few techniques have high prevalence, while the others have low or very low frequencies [24].

One scheme for dealing with class imbalance is oversampling [24]. This data-level approach consists of randomly oversampling duplicate examples from low-frequency classes to rebalance the class distribution. However, this can result in overfitting and we opted to use the TextAttack Framework [25] for generating adversarial examples. TextAttack is a Python framework designed for adversarial attacks, data augmentation, and adversarial training in NLP. The adversarial attack finds a sequence of transformations to perform on an input text such that the perturbations adhere to a set of grammar and semantic constraints and the attack is successful [26]. These transformations performed can be reused to expand the training dataset by producing perturbed versions of the existing samples. As such, TextAttack Framework offers various pre-packaged recipes for data augmentation [27].

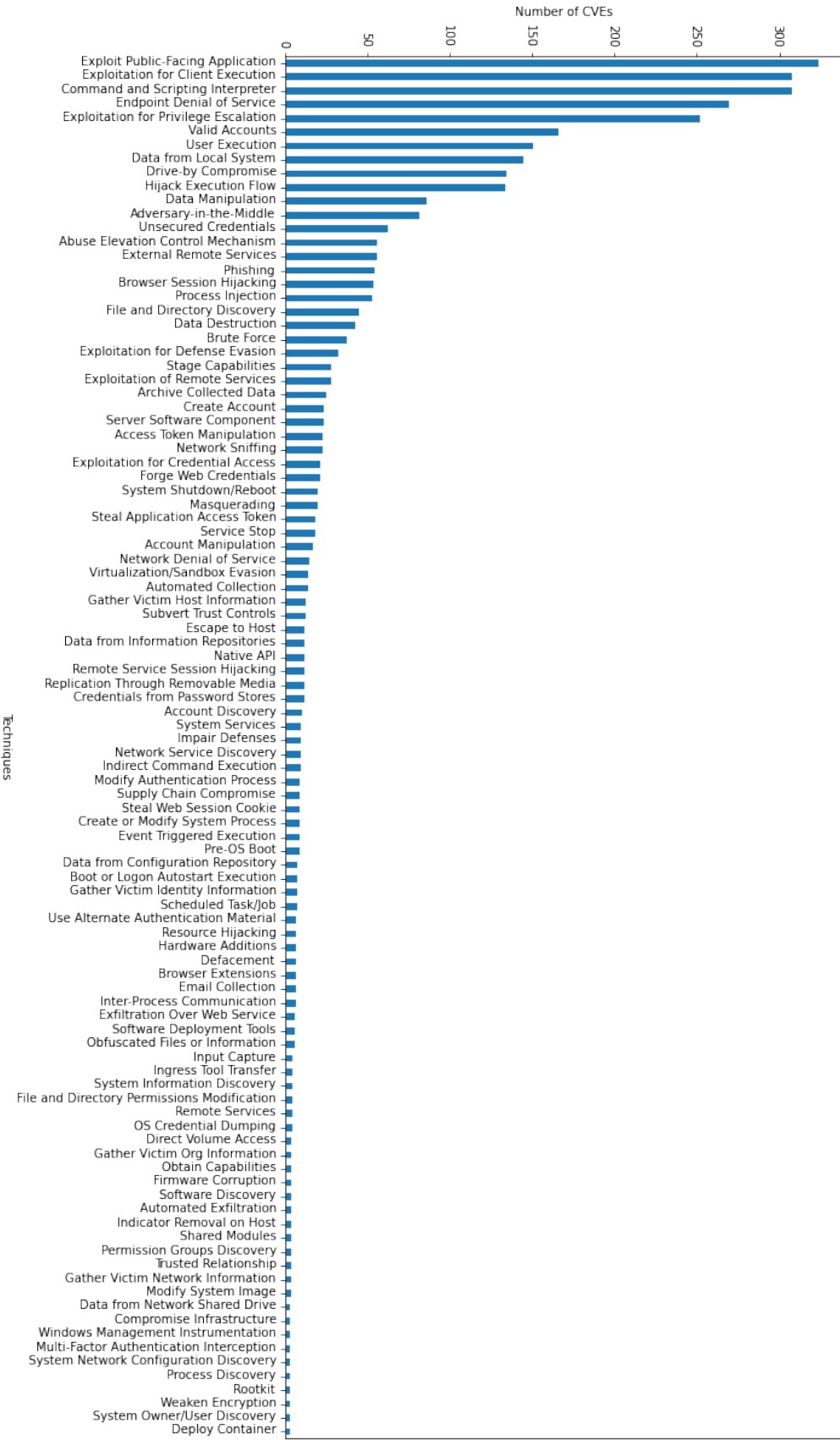

**Figure 1.** The distribution of CVEs among techniques.

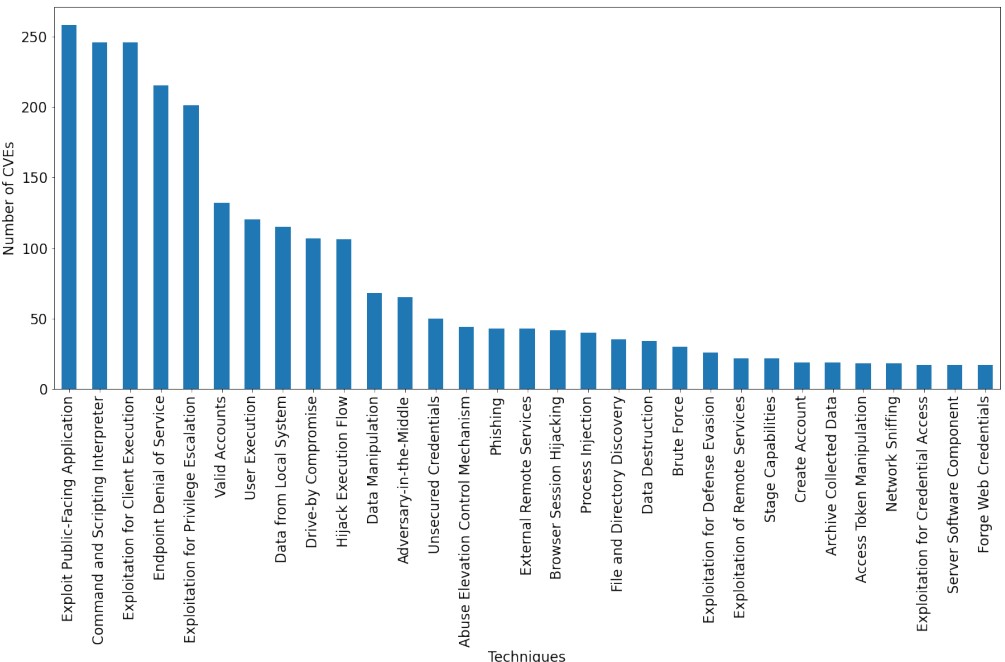

**Figure 2.** The distribution of CVEs among the 31 considered techniques after applying the threshold.

We chose the EasyDataAugmenter (EDA) for augmenting the CVE dataset, which performs four simple but powerful operations on the input texts: synonym replacement, random insertion, random swap, and random deletion. EDA significantly boosts performance and shows particularly strong results for smaller datasets [28], which makes it the perfect candidate for oversampling our labeled CVE corpus. Moreover, EDA does not perform major alterations of the content and is not as computationally expensive as other recipes, such as CLAREAugmenter, while providing satisfactory results on our CVE corpus.

Since one CVE can be mapped to multiple techniques at the same time, rare techniques among the dataset are usually found in combination with highly prevalent techniques. Using all CVEs that are mapped to a specific technique for augmentation would only preserve the class imbalance, generating new samples for both low-frequency and high-frequency techniques. To counter this undesired effect, EasyDataAugumenter was fed only with CVEs that were particular to only one technique and were mapped to that technique only, thus producing new samples only for the desired class.

Figure 3 displays the distribution of CVEs per technique after performing the data augmentation. The initial severe imbalance among techniques was scaled down, but still exists, due to the reduced number of particular CVEs for low-frequency techniques.

## 2.2. Machine Learning and Neural Architectures

Our main goal is to create a model that can accurately predict all the techniques that can be mapped to a specific CVE while using its text description. We tacked this task as a multi-label learning problem as each CVE may be assigned to a subset of techniques. Given the challenging nature of the multi-label paradigm [29], we experimented with multiple state-of-the-art machine learning models to find the most predictive architecture.

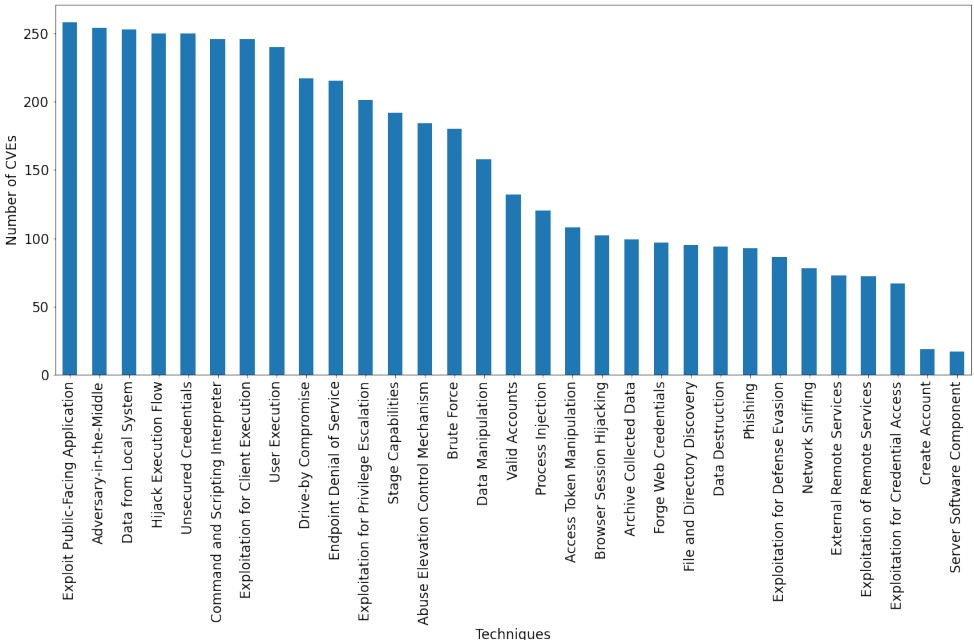

**Figure 3.** The distribution of CVEs among the 31 considered techniques after data augmentation.

### 2.2.1. Classical Machine Learning

In order to establish a strong baseline we also considered classical machine learning algorithms applied on bag-of-words representations. All CVE descriptions were preprocessed to remove noise and retain only the relevant words. The pipeline from the spaCy [30] NLP open-source library was employed which included the following steps: text tokenization, removal of stopwords, punctuation, and numbers, followed by lemmatization of remaining tokens. The tokens are afterward converted to bag-of-words representations using Term Frequency-Inverse Document Frequency (TF-IDF).

#### Multi-Label Learning

The aim of problem transformation methods is to reduce the complexity of the multi-label learning by converting the multi-label problem into one or more single-label classification tasks [31].

Given that the interconnection between techniques is worth taking into account when labelling a CVE since it can provide further insights on general adversarial patterns, we experimented with different problem transformation methods to find the one that captures best the relations between labels:

- *One versus Rest*. This method splits the multi-label problem into multiple binary classification tasks, one for each label, treated independently. The *N* different binary classifiers are separately trained to distinguish the examples of a single class from all the examples from the other labels [32];
- *Label Powerset*. This method considers every unique combination of labels as a single class, reducing the multi-label problem to a multi-class classification problem [29]. The real advantage of this strategy is that correlations between labels are exploited for a more accurate labelling process;
- *Binary Relevance*. This linear strategy groups all positive and negative examples within a label into a set, later training a classifier for each resulted set. The final prediction is then computed by merging all the intermediary predictions of the trained classifiers [29]. An advantage of this strategy consists of the possibility to perform parallel executions;
- *RaKEL(Random k-Labelsets)*. This state-of-the-art approach builds an ensemble of Label Powerset classifiers trained on a different subset of the labels [33].

Naive Bayes Classifiers

The Naive Bayes classifier makes the simplifying assumption that features are conditionally independent, given a class. Even though the assumption of independence is generally unrealistic, Naive Bayes performs well in practice, competing with more sophisticated classifiers models especially for text classification [34]. We chose to experiment with a Naive Bayes variant for multinomial distributed data because of the model's simplicity and relatively good results.

Support Vector Machines

A Support Vector Machine (SVM) searches for the maximum margin hyperplane that separates two classes of examples. Because SVMs have shown efficiency to capture high dimensional spaces and performed successfully on a number of distinctive classification tasks [35], we decided to use it in our experiments for CVE technique labelling. We performed an exhaustive search over specified parameters values using GridSearchCV [36] to determine the optimum configuration of parameters.

### 2.2.2. Convolutional Neural Network (CNN) with Word2Vec

Convolutional Neural Networks (CNNs) consist of multiple layers designed to extract local features in the form of a feature map. Since CNN uses back-propagation to update its weights in the convolutional layers, the CNN feature extractors are self-determined through continuous tuning of the model [37]. In the field of NLP, CNNs have proved to be extremely effective in several tasks, such as semantic parsing [38] and sentence modeling [39]. This intuition pointed in the direction to experiment with CNN for our model since CNNs with Word2Vec embeddings are robust even on small datasets. In addition, we considered SecVuln_WE [40] that includes word representation especially designed for the cybersecurity vulnerability domain. SecVuln_WE was trained on security-related sources such as Vulners, English Wikipedia (Security category), Information Security Stack Exchange Q&As, Common Weakness Enumeration (CWE) and Stack Overflow.

Figure 4 presents the architecture in which the pre-trained SecVuln_WE embeddings are passed through the convolutional layer containing 100 filters with a kernel size of 4. In this way, each convolution will consider a window of 4 word embeddings. Afterward, we perform batch normalization of the activations of the previous layer at each batch. Next comes the MaxPool and the Dropout layers, followed by a dense layer with sigmoid activation. Since we are dealing with a multi-label classification problem, the output layer has a designated node for each technique and each output indicates the binary probability to have a specific technique mapped to the considered CVE.

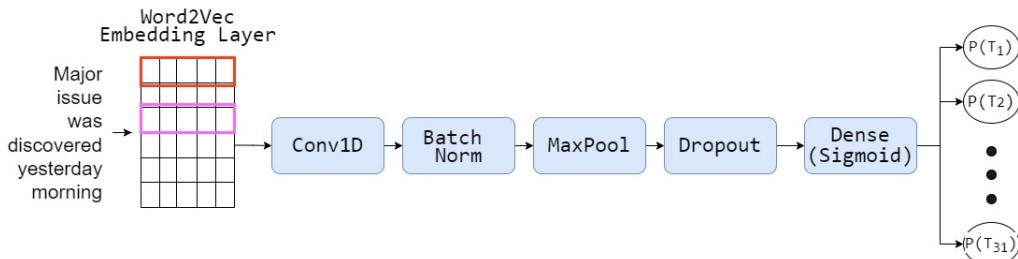

**Figure 4.** Architecture of the CNN with Word2Vec embeddings.

### 2.2.3. BERT-Based Architecture with Multiple Output Layers

Reducing the considerable complexity of the multi-label problem was first among our considerations when designing this architecture. Converting our multi-labeling problem into multiple binary classification tasks following the *One versus Rest* method has the advantage of conceptual simplicity; yet, having a distinct BERT layer for contextualized embeddings for each one of the 31 techniques was redundant.

The proposed architecture from Figure 5 considers a pre-trained BERT encoder, a Dropout layer, and an individual dense layer for each technique, which outputs the probability that a particular CVE points to that particular technique. The model is consistent with the considerations of the *One VS Rest* method, while also taking advantage of the shared embeddings layer.

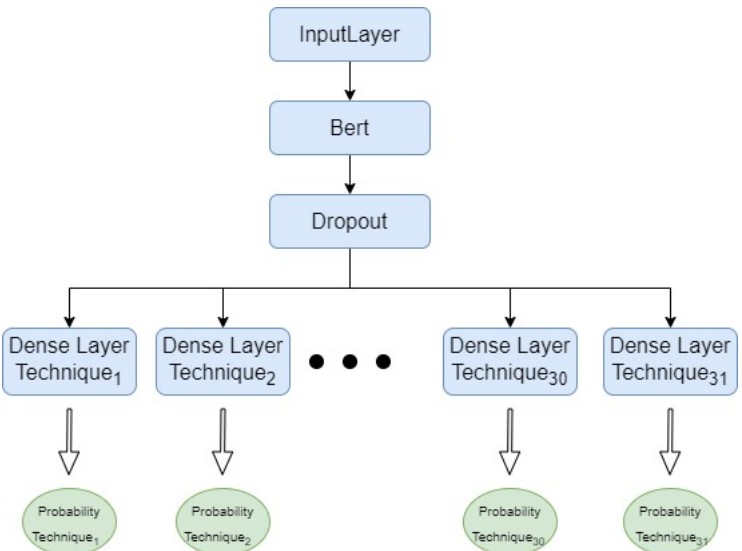

**Figure 5.** BERT-based architecture with multiple output layers.

### 2.2.4. BERT-Based Architecture Adapted for Multi-Labeling

Analyzing each label separately might overlook the strong correlation between techniques. This correspondence has multiple roots, as techniques in a given tactic are connected through their attack behavior pattern, whereas techniques across multiple tactics are connected through the attack vector of the vulnerability. Thus, we explored creating a model capable of exploiting the link between multiple techniques.

The specific architectural decision taken for this last design was to have only one output layer, with one individual node for each technique. In this manner, we aim to capture the specifics for each technique, while also considering how subsets of techniques are interconnected.

Figure 6 details the proposed model which considers 768-dimensional contextual embeddings from various BERT-based models (i.e., BERT [41], SciBERT [42], and SecBERT [43]) passed through a Dropout layer. The Dropout layer output goes through a Linear layer with 768 input features and 31 output nodes, one for each technique. We considered BCEWith-LogitsLoss [44] (the combination of a Sigmoid layer and the BCELoss) as a loss function, the most commonly used for multi-label classification tasks, because each output node reveals the probability of a technique to be tagged for a specific CVE (i.e., the probabilities need to be treated independently).

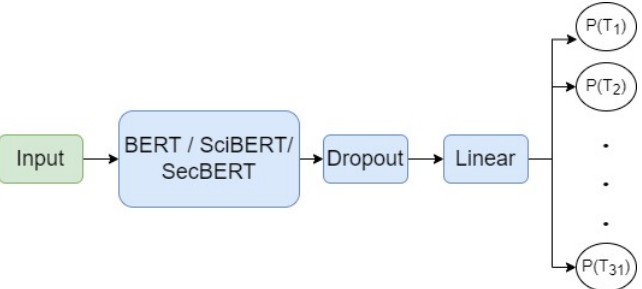

**Figure 6.** The design of the multi-labeling BERT-based architecture.

*2.3. Performance Assessment*

For a predicted technique, we wanted to make sure that our mapping was correct (i.e., high precision—P) and we wanted to correctly classify as many examples as possible for a given class (i.e., high recall—R). Thus, we considered the F1-score as a performance metric for all models, defined as the harmonic mean of the P and R per class. Moreover, we used the weighted version of the F1-score given the imbalance between classes, which calculated a general F1-score per model by proportionally combining the F1-scores obtained for each label separately. We also computed the weighted precision and recall for the tested models.

## 3. Results

This section analyses the results of the empirical experiments performed using the previously detailed models. First, it compares the performance of various models. Second, it assesses the impact of data augmentation on performance and investigates the metrics obtained by the best model.

Multiple observations can be made based on the results of our experiments shown in Table 1. From the classical machine learning models, LabelPowerset is the best multi-label strategy and SVC with a linear kernel and C = 32 has the higher F1-score, competing even with our deep-learning models. The SecBERT model has the highest F1-score (42.34%) among all considered models, proving to be the most powerful solution to labeling a CVE. An important observation is that the CNN + Word2Vec architecture obtained better results than those using simple BERT. Thus, domain-related pre-training on large security databases leads to increased performance by providing better contextualization and partially compensating for the scarce training set.

**Table 1.** Results for the proposed models (italics marks the best multi-label strategy for classical ML, while bold marks the best model).

| Model Type | Model | Multi-Label Strategy | Weighed P | Weighed R | Weighed F1-Score |
|---|---|---|---|---|---|
| Classical ML | Naive Bayes | OneVsRestClassifier | *57.35%* | 9.18% | 14.47% |
| | | LabelPowerset | 31.40% | 24.59% | 24.76% |
| | | BinaryRelevance | 57.35% | 9.18% | 14.47% |
| | | RakelD | 53.71% | 9.83% | 15.31% |
| | SVC | OneVsRestClassifier | 31.97% | *35.57%* | 33.32% |
| | | LabelPowerset | *46.73%* | 34.75% | *37.98%* |
| | | BinaryRelevance | 33.45% | 34.91% | 33.75% |
| | | RakelD | 36.20% | 33.77% | 34.50% |
| Deep Learning | CNN + Word2Vec | - | 48.32% | **35.40%** | 39.39% |
| | Multi-Output BERT | - | 46.85% | 31.47% | 35.92% |
| | Multi-label BERT | - | 55.25% | 30.98% | 37.43% |
| | Multi-label SciBERT | - | **59.26%** | 34.42% | 41.87% |
| | Multi-label SecBERT | - | 57.66% | **35.40%** | **42.34%** |

Table 2 points out the appropriateness of employing data augmentation techniques on our dataset for deep learning models (approximately 6% performance gain). Only the best multi-label strategy for classical machine learning algorithms was considered. The F1-score falls considerably by 10% for Naive Bayes, in particular, since Naive Bayes places great importance on the number of appearances of a word in a document; however, swapping a relevant word with synonyms and performing random insertions or deletions (i.e., the strategies employed by the EasyDataAugmenter [28]) only confuse the model. The SVC model had a similar performance, whereas the BERT-based models take advantage of the increased sample size/the decreased class imbalance, and generalize better. Not only is performance increased, but the models also tend to learn faster (see faster convergence in Figure 7 in terms of training loss for each output layer associated with a technique in the multi-output BERT model). Moreover, Figure 7 denotes which techniques are more easily learned by the model.

**Table 2.** Side-by-side comparison of performance with and without data augmentation (bold denotes the best model).

| Model | Data Augmentation | Weighted P | Weighted R | Weighted F1-Score |
|---|---|---|---|---|
| Naive Bayes (LabelPowerset) | No | 31.40% | 24.59% | 24.76% |
| | Yes | 29.40% | 14.42% | 14.42% |
| SVC (LabelPowerset) | No | 46.73% | 34.75% | 37.98% |
| | Yes | 45.90% | 34.09% | 36.79% |
| CNN + Word2Vec | No | 48.32% | 35.40% | 39.39% |
| | Yes | 50.48% | 35.59% | 41.59% |
| Multi-Output BERT | No | 46.85% | 31.47% | 35.92% |
| | Yes | 49.81% | 35.57% | 39.66% |
| Multi-label SciBERT | No | **59.26%** | 34.42% | 41.87% |
| | Yes | 52.52% | **45.90%** | **47.84%** |
| Multi-label SecBERT | No | 57.66% | 35.40% | 42.34% |
| | Yes | 54.70% | 42.45% | 46.54% |

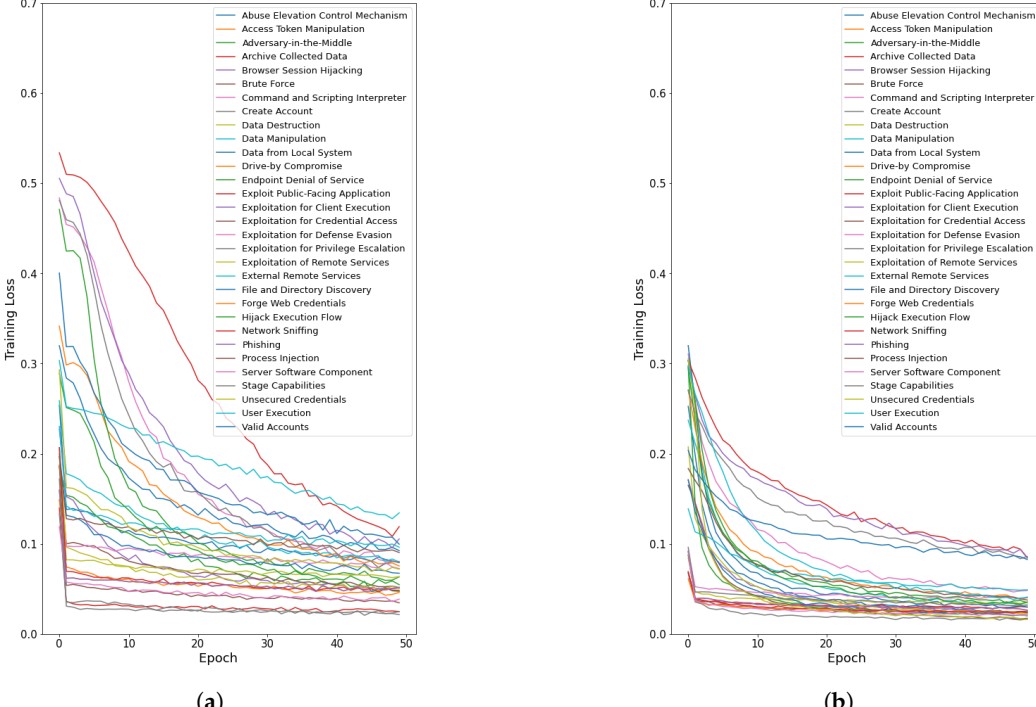

(**a**)             (**b**)

**Figure 7.** Comparison of training loss for the multi-output BERT architecture. (**a**) Without data augmentation; (**b**) With data augmentation.

Since Table 2 only provides a global overview of the average performance of the SciBERT model trained on the augmented data, exploring the particular difference between how the model handles different techniques provides additional insights into our model's behavior. Figure 8 plots the F1-score obtained for each individual technique, for both the original model and the one trained on the augmented dataset. Apart from four exceptions (*Data from Local System*, *Hijack Execution Flow*, *User Execution* and *File and Directory Discovery*), the model obtains considerably higher or at least equal scores for all the other 27 techniques. Moreover, the difference between models is minimal (close to 0) for the techniques where the initial model obtains a better F1-score.

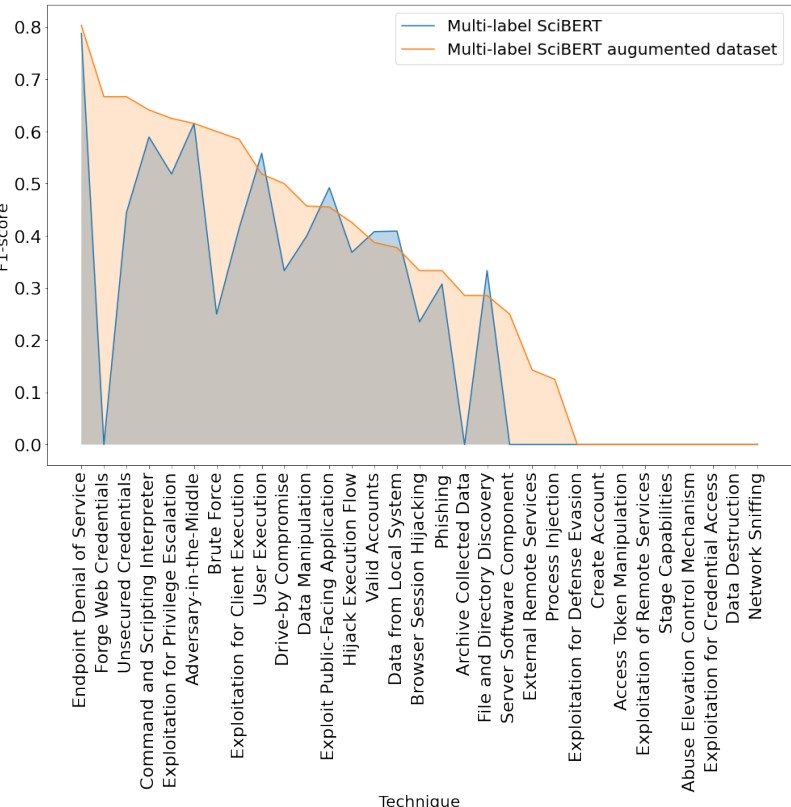

**Figure 8.** Comparing F1-score per technique between SciBERT model trained on initial and augmented dataset.

The added gain of the multi-label SciBERT model trained on the augmented dataset resides in its ability to maximize the F1-score for techniques where the initial model performed poorly. One such example is *Forge Web Credentials*. The initial model obtained an F1-score of 0% since both recall and precision were 0%. However, the improved version of the model obtained an F1-score of 66.66%, with a recall of 50% and precision of 100% after data augmentation; similarly, data augmentation tuned the model to predict the *Forge Web Credentials* technique with 100% precision. Overall, the number of techniques with which the model had difficulty in learning has decreased substantially.

Figure 9 shows the correlation between the CVE distribution and the F1-score obtained for the SciBERT models, both using the initial dataset and the one trained after augmentation. The techniques are displayed on both graphs in the same order to indicate how the CVE distribution changed after performing the process of data augmentation and how the adjustments in CVE distribution impacted the F1-score. We observe that not only the techniques initially associated with a small number of CVEs benefited from the augmentation method, but also the techniques associated with a high distribution of samples—for example, the F1-score for the *Command and Scripting Interpreter* technique increased from the initial 58.92% to 64.12%.

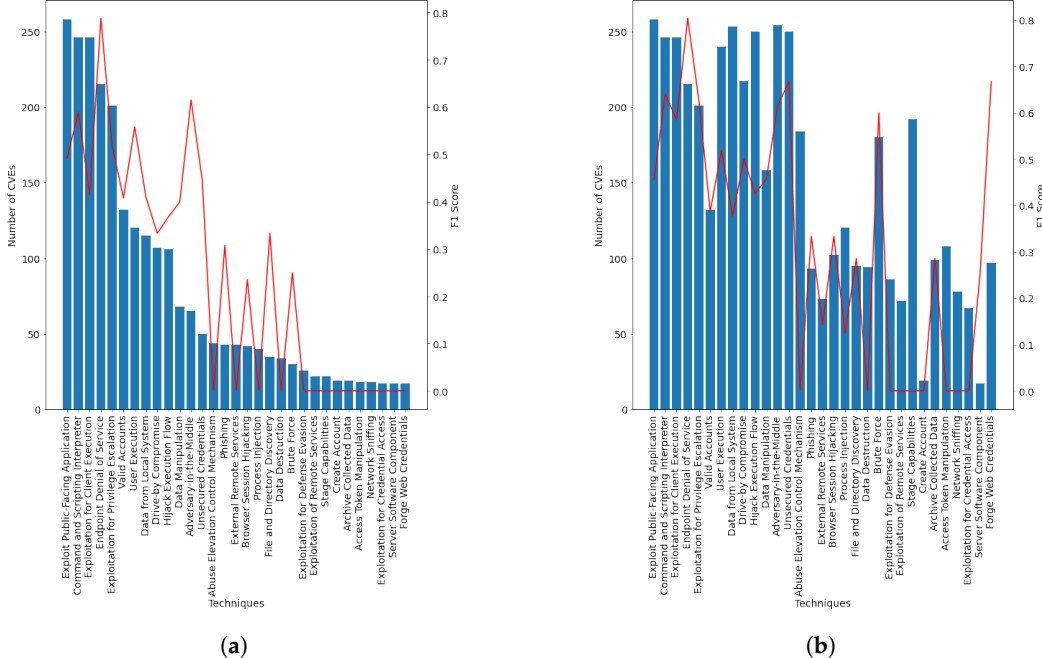

|     (a)     |     (b)     |

**Figure 9.** Comparing the F1-score over the CVE distribution for the SciBERT model. (**a**) Without augmentation; (**b**) With augmentation.

**Table 3.** Precision, Recall and F1-Scores for the best model.

| Technique | Weighted P | Weighted R | Weighted F1-Score |
|---|---|---|---|
| Endpoint Denial of Service | 77.58% | 83.33% | 80.35% |
| Forge Web Credentials | 100.00% | 50.00% | 66.66% |
| Unsecured Credentials | 60.00% | 75.00% | 66.66% |
| Command and Scripting Interpreter | 60.00% | 68.85% | 64.12% |
| Exploitation for Privilege Escalation | 56.45% | 70.00% | 62.50% |
| Adversary-in-the-Middle | 80.00% | 50.00% | 61.53% |
| Brute Force | 100.00% | 42.85% | 60.00% |
| Exploitation for Client Execution | 50.87% | 50.81% | 58.49% |
| User Execution | 58.33% | 46.67% | 51.85% |
| Drive-by Compromise | 64.70% | 40.74% | 50.00% |
| Data Manipulation | 44.44% | 47.05% | 45.71% |
| Exploit Public-Facing Application | 48.27% | 43.07% | 45.52% |
| Hijack Execution Flow | 50.00% | 37.03% | 42.55% |
| Valid Accounts | 41.37% | 36.36% | 38.70% |
| Data from Local System | 41.66% | 34.48% | 37.73% |
| Browser Session Hijacking | 42.85% | 27.27% | 33.33% |
| Phishing | 42.85% | 27.27% | 33.33% |
| Archive Collected Data | 50.00% | 20.00% | 28.57% |
| File and Directory Discovery | 40.00% | 22.22% | 28.57% |
| Server Software Component | 50.00% | 16.66% | 25.00% |
| External Remote Services | 50.00% | 8.33% | 14.28% |
| Process Injection | 25.00% | 8.33% | 12.50% |
| *Exploitation for Defense Evasion (26)* | 0.00% | 0.00% | 0.00% |
| *Create Account (19)* | 0.00% | 0.00% | 0.00% |
| *Access Token Manipulation(18)* | 0.00% | 0.00% | 0.00% |
| *Exploitation of Remote Services (22)* | 0.00% | 0.00% | 0.00% |
| *Stage Capabilities (22)* | 0.00% | 0.00% | 0.00% |
| *Abuse Elevation Control Mechanism (44)* | 0.00% | 0.00% | 0.00% |
| *Exploitation for Credential Access (17)* | 0.00% | 0.00% | 0.00% |
| *Data Destruction (34)* | 0.00% | 0.00% | 0.00% |
| *Network Sniffing (18)* | 0.00% | 0.00% | 0.00% |

## 4. Discussion

### 4.1. In-Depth Analysis of the Best Model

Table 3 introduces a complete overview of the results recorded for the best model, the multi-label SciBERT trained on the augmented dataset. The F1-score per technique from the MITRE ATT&CK Enterprise Matrix ranges from 80.35% for *Endpoint Denial of Service* to 0.00%; the last techniques at the end of Table 3 marked with italics and including the corresponding number of training samples in parenthesis. Even though the model scores on a global scale an F1-score of 47.84%, the model fails to capture any knowledge about nine out of the thirty-one techniques, though fewer instances than the other evaluated models. We can associate this inability of the model to recognise the distinct features of these techniques with the extremely reduced number of samples for each technique, even after performing data augmentation. The existing samples in the dataset do not contain enough relevant characteristics for these techniques; as such, the model cannot differentiate them.

Nevertheless, the model successfully captures the essence of other techniques, obtaining a precision of 100.00% for *Forge Web Credentials* and *Brute Force*. For almost all techniques, precision exceeds recall, thus indicating that the general tendency of the model is to omit a label, rather than misplace a technique that cannot be mapped to a particular CVE.

Overall, given the complexity of the multi-label problem and the severe imbalance of the training set, the model obtains promising performance for a subset of techniques, while managing to maximize its overall F1-score.

### 4.2. Error Analysis

This subsection revolves around understanding the roots of the multi-label SciBERT model limitation. After a methodological investigation that aims to identify the cause of the model's errors, the observed performance deficiencies are further discussed.

Table 4 presents different CVEs whose predicted techniques differ partially or completely from the labeled ones. For most errors in the dataset with multiple techniques tagged, the model succeeds in labeling a subset of correct techniques. This observation stands true for errors 1, 2, and 3 from Table 4. While analyzing error #1, the model extracts the most obvious technique, pointed out by language markers such as *password unencrypted*, *global file*, but fails to make the deduction that, in order for a user to access the file system, a valid account must be used. In contrast, the model successfully identifies the *Valid Accounts* technique for error #2. In general, techniques that are not clearly textually encapsulated and whose understanding requires prerequisite knowledge are overlooked by the model.

Figure 10 studies the model's choice of labels for CVE #2 from Table 4 using Lime [18], the model successfully recognizes the predominant label (i.e., *Valid Accounts*). Moreover, the model correctly identifies the most important concept, the word *authenticated*, which points in the direction of *Valid Accounts*. We can observe that there are techniques that are not ambiguous for the model and for which the labeling process is straightforward; such an example is *Valid Accounts*. The model extracts only the relevant features for the label and the technique is correctly identified. For the *Exploitation for Client Execution*, the model identifies patterns that suggest that the CVE should be mapped to the given technique, as well as patterns that suggest the contrary. Being capable to identify features that are correlated to both situations confuses the model. This problem results from the fact that the meaning behind multiple techniques is overlapping and, as a result, relevant features for a given technique cannot be differentiated.

**Table 4.** Comparing predictions with the true values for the best model.

| # | CVE Text | True Techniques | Predicted Techniques |
|---|----------|-----------------|----------------------|
| 1 | Jenkins Publish stores password unencrypted in its global configuration file on the Jenkins controller where it can be viewed by users with access to the Jenkins controller file system. | Unsecured Credentials, Valid Accounts | Unsecured Credentials |
| 2 | Due to improper input validation in InfraBox, logs can be modified by an authenticated user. | Valid Accounts, Exploitation for Client Execution | Valid Accounts |
| 3 | In Django 2.2 MultiPartParser, UploadedFile, and FieldFile allowed directory traversal via uploaded files with suitably crafted file name | File and Directory Discovery, Command and Scripting Interpreter | File and Directory Discovery, Exploit Public-Facing Application |
| 4 | Whale browser for iOS before 1.14.0 has an inconsistent user interface issue that allows an attacker to obfuscate the address bar which may lead to address bar spoofing. | Browser Session Hijacking | User Execution |
| 5 | isula-build before 0.9.5-6 can cause a program crash, when building container images, part of the functions for processing external data do not remove spaces when processing data. | Exploitation for Client Execution | Endpoint Denial of Service |

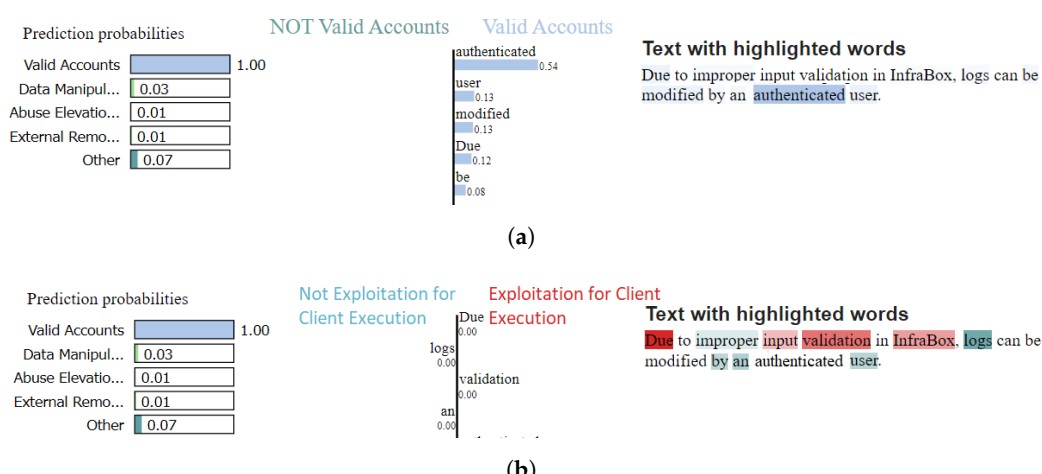

**Figure 10.** Comparison of word mappings for each technique corresponding to CVE #2 from Table 4. (**a**) Mapping Valid Accounts; (**b**) Mapping Exploitation for Client Execution.

An interesting aspect is revealed in error #3, namely that the model correctly tags *File and Directory Discovery*, but also associates the CVE with *Exploit Public Facing Application*, instead of *Command and Scripting Interpreter*. Both techniques in the MITRE ATT&CK Enterprise Matrix could be equally correctly mapped on the given text description. This is an important observation and points out the established CVE labeling methodology; this highlights a fault in the data collection procedure, rather than the model's capacity to learn the multi-labeling problem. Example #4 presents a similar case, since the predicted technique *Endpoint Denial of Service* is a correct label for the CVE, although it does not appear among the true labels.

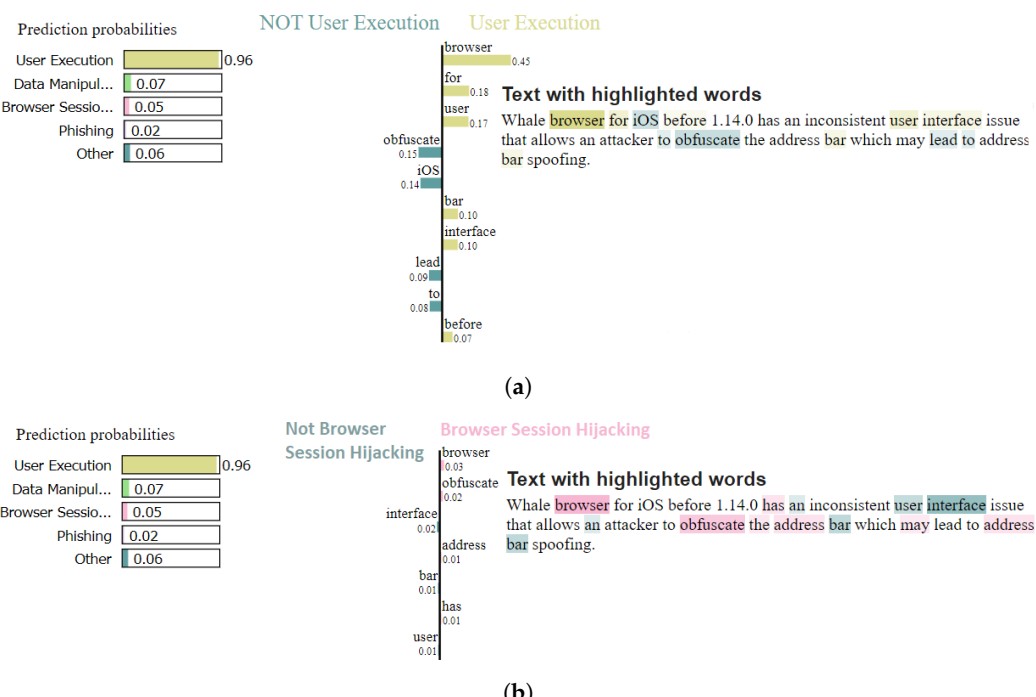

**Figure 11.** Comparison of word mappings for each technique corresponding to CVE #4 from Table 4. (**a**) Mapping User Execution technique; (**b**) Mapping Browser Session hijacking.

Error #4 is analyzed in detail in Figure 11 to observe insights on how the model associates the features. The word *browser* is highlighted for both the predicted and the correct label. However, the difference resides in the relevance percentage associated with the word for each label, namely 0.45 for *User Execution* and 0.03 for *Browser Session Hijacking*. While the word *browser* is recognized as being relevant for both labels, the label with the higher percentage is selected. This finding can be associated with the discrepancy between training examples—240 for *User Execution*, while *Browser Session Hijacking* has only 102. Thus, the class imbalance affects the model's capability to recognize the real correlation between features and techniques, and leads the model to a biased decision.

The model extracts a correct technique for error #5 in Table 4, although it was not among the true labels. As Figure 12 shows, the CVE text description indicates the *Endpoint Denial of Service* technique, since the word *crash* is present and the relevance of the word for the *Endpoint Denial of Service* technique is 0.93. Figure 12 also suggests that the word *crash* is the only word that has a high impact on the model's decision to label the CVE as *Endpoint Denial of Service*.

Two observations can be made based on Figure 12. One is that the model successfully captures a technique overlooked by the reviewer. The technique labeling process is error-prone due to the ambiguity of the CVE text description and also the complexity of the labeling processing given the wide range of available techniques. Second, the model assigns a higher relevance to features that suggest *Endpoint Denial of Service* even though key features for the *Exploitation for Client Execution* are identified (i.e., *program* and *functions*).

Table 5 presents the most relevant words when performing feature extraction for each technique. More than 50% of the techniques have the same most relevant feature in common with other techniques in the MITRE ATT&CK Enterprise Matrix. For example, *Exploitation for Privilege Escalation*, *Data from Local System*, *Data Destruction*, *Browser Session Hijacking*, *Archive Collected Data*, and *Create Account* are all mapped to the same feature. Having the same most relevant extracted feature implies a strong intersection between techniques. This further emphasizes that the separation between labels is fuzzy. The opinion and consensus among reviewers were used to separate ambiguous examples, making use of previous experience and context obtained from other resources. This is inherited by the model since the labels from the training set reflect the reviewers' perspective. In this context, more

information would be valuable to counter the bias encapsulated in the training set by offering more background information to the model.

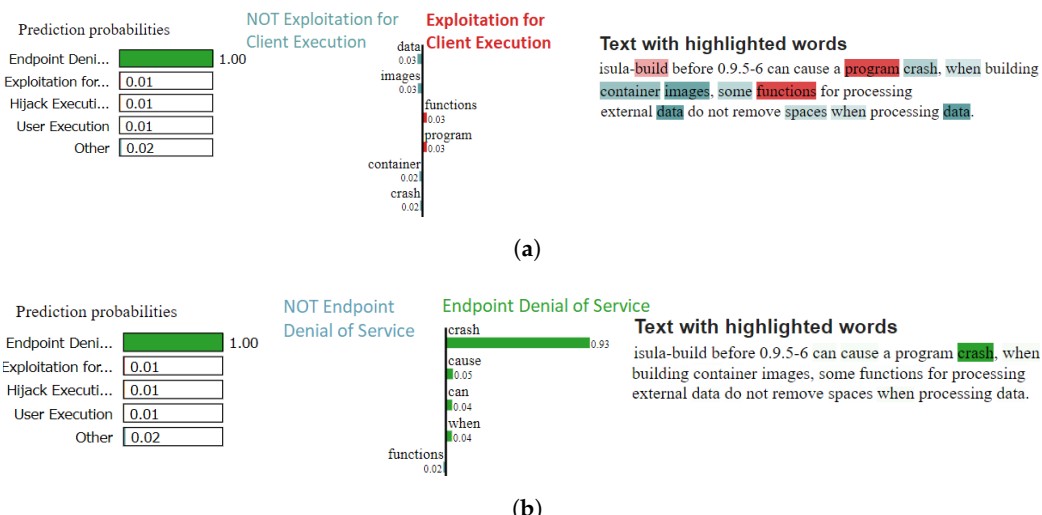

**Figure 12.** Comparison of word mappings for each technique corresponding to CVE #5 from Table 4. (**a**) Mapping Exploitation for Client Execution; (**b**) Mapping Endpoint Denial of Service.

**Table 5.** The most important words extracted per technique.

| Technique | CVEs |
| --- | --- |
| Exploitation for Privilege Escalation | arbitrary |
| Data from Local System | arbitrary |
| Data Destruction | arbitrary |
| Browser Session Hijacking | arbitrary |
| Archive Collected Data | arbitrary |
| Create Account | arbitrary |
| Forge Web Credentials | bypass |
| Unsecured Credentials | bypass |
| External Remote Services | bypass |
| Adversary-in-the-Middle | trigger |
| Phishing | trigger |
| Stage Capabilities | trigger |
| Exploitation for Credential Access | wordpress |
| Brute Force | wordpress |
| Abuse Elevation Control Mechanism | xml |
| Endpoint Denial of Service | parameter |
| Network Sniffing | parameter |
| User Execution | remote |
| Drive-by Compromise | remote |
| Server Software Component | service |
| Data Manipulation | service |
| Exploit Public-Facing Application | version |
| Command and Scripting Interpreter | pointer |
| Exploitation for Client Execution | attack |
| Valid Accounts | system |
| Hijack Execution Flow | cause |
| Process Injection | privilege |
| File and Directory Discovery | execute |
| Exploitation for Defense Evasion | use |
| Exploitation of Remote Services | possibly |
| Access Token Manipulation | header |

*4.3. Limitations*

We have identified a number of limitations for our model, which have a toll on the model's performance; these limitations are detailed further. First, the process of manually labeling a CVE is inevitably affected by the subjective perspective of the reviewer. Even though multiple attempts to limit this undesired outcome were taken (i.e., following a clear methodology and establishing general guidelines for the reviewers), the annotators were unable to fully eliminate the inconsistency in the dataset labels.

Second, the quality of the information in the CVE text descriptions must also be taken into consideration when discussing the general limitations of the proposed model. Inconsistencies among the CVE descriptions (incomplete, outdated, or even erroneous details) are highly prevalent [45], thus narrowing the attainable performance of the model.

Third, there is no clear delimitation between certain techniques. Multiple techniques have overlapping meanings and follow the same attack pattern (e.g., *Exploitation for Defence Evasion* and *Abuse Elevation Control Mechanism*). Due to this, a CVE might have multiple possible correct labels, depending on the methodology used to mark the CVE since techniques are closely interconnected and the difference between relating techniques is generally subtle.

Lastly, the rather small dataset and the severe imbalance between the number of CVEs associated with a technique has a toll on the capacity of the model to accumulate enough knowledge to correctly label future samples. Having a larger knowledge base for training the model would help provide samples so that the model perceives also sensitive nuances in CVE text descriptions.

## 5. Conclusions

In this paper, we emphasized the need for an automatic linkage between the CVE list and MITRE ATT&CK Enterprise Matrix techniques. The problem was transposed into a multi-label task for Natural Language Processing for which we introduce a novel labeled CVE corpus that was augmented using adversarial attacks to limit the severe impact of imbalance between labels. Our baseline includes several classic machine learning models and BERT-based architectures, and the best performing model (i.e., Multi-label SciBERT) was evaluated within a series of experiments from multiple perspectives to extract a complete overview of the data augmentation impact. Comparing the obtained metrics against classical machine learning models accentuates the significant benefits brought by our solution to labeling CVEs with corresponding techniques.

Despite our model obtaining promising results in terms of well-represented techniques, the inherent limitations imposed by the training set tops up the maximum achievable performance. Future work will focus on improving the robustness of the labeled CVE corpus. On one hand, we will focus on enforcing homogeneity among labeling methodology; on the other, we will address the severe imbalance between labels and also its reduced size. Possible new strategies might consider Few-Shot Learning methods [46] for task generalization considering few samples. Semi-supervised learning [47] could also be a possible research direction, given the reduced number of labeled CVEs and the significant number of unlabeled samples that exist in the CVE list. Another aspect that is worth exploring is whether or not gathering extra information from additional sources (e.g., *Common Weakness Enumeration* CWE [48]) can address the incompleteness and inconsistency of the textual CVE description.

**Author Contributions:** Conceptualization, O.G., A.N., M.D. and R.R.; methodology, O.G. and M.D.; software, A.N. and O.G.; validation, O.G., A.N. and M.D.; formal analysis, O.G., A.N. and M.D.; investigation, A.N., O.G. and M.D.; resources, O.G. and A.N.; data curation, A.N.; writing—original draft preparation, O.G. and A.N.; writing—review and editing, M.D. and R.R.; visualization, A.N.; supervision, M.D. and R.R.; project administration, R.R.; funding acquisition, R.R. All authors have read and agreed to the published version of the manuscript.

**Funding:** This work was supported by a grant of the Romanian National Authority for Scientific Research and Innovation, CNCS—UEFISCDI, project number 2PTE2020, YGGDRASIL—"Automated System for Early Detection of Cyber Security Vulnerabilities".

**Institutional Review Board Statement:** The study was conducted in accordance with the Declaration of Helsinki, and approved by the Ethics Committee of the Faculty of Automated Control and Computers, University Politehnica of Bucharest.

**Informed Consent Statement:** Not applicable.

**Data Availability Statement:** The dataset is freely available on Tagtog at https://www.tagtog.com/readerbench/MitreMatrix/ (accessed on 8 August 2022), whereas the code is available on Github at https://github.com/readerbench/CVE2ATT-CK (accessed on 8 August 2022).

**Acknowledgments:** We would also like to show our gratitude to Ioana Nedelcu, Ciprian Stanila, and Ioana Branescu for their contributions to building the labeled CVE corpus.

**Conflicts of Interest:** The authors declare no conflict of interest.

## Abbreviations

The following abbreviations are used in this manuscript:

| | |
|---|---|
| ATT&CK | Adversarial Tactics, Techniques, and Common Knowledge |
| BERT | Bidirectional Encoder Representations from Transformers |
| CAPEC | Common Attack Pattern Enumeration and Classification |
| CNN | Convolutional Neural Network |
| CVE | Common Vulnerabilities and Exposures |
| CVET | Common Vulnerabilities and Exposures Transformer |
| CWE | Common Weakness Enumeration |
| EDA | Easy Data Augmentation |
| ML | Machine Learning |
| NLP | Natural Language Processing |
| SciBERT | Scientific Bidirectional Encoder Representations from Transformers |
| SecBERT | Security Bidirectional Encoder Representations from Transformers |
| SVM | Support Vector Machine |
| TF-IDF | Term Frequency-Inverse Document Frequency |
| TRAM | Threat Report ATT&CK Mapping |

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
