# Peer review of "CVE2ATT&CK: BERT-Based Mapping of CVEs to MITRE ATT&CK Techniques"

_algorithms, doi:10.3390/a15090314_

Round 1

Reviewer 1 Report

The study presents an automatic approach relating CVE repositories with MITRE attack matrix based on standard architectures and tools in NLP. A significant amount of work went into this study. Well described, clear presentation - all in all a nice piece of work with relevance for cybersecurity in practice.

Author Response

Thank you kindly for your comments, we highly appreciate it!

Reviewer 2 Report

1. The following project should be mentioned in the manuscript and the authors should clarify the positioning of the following project and the authors' research.

https://medium.com/mitre-engenuity/cve-mitre-att-ck-to-understand-vulnerability-impact-c40165111bf7

2. The manuscript does not sufficiently explain the specific benefits of linking entries in the two databases. The authors should provide specific examples of entries and links, and explain the condition for linking and what the practical benefits of such a linking result are.

3.  Isn't it sufficient to simply have tags added when adding a new entry to CVE? The authors should explain why the proposed machine learning-based approach is superior to the method of tagging CVE entries when they are added or the method of community-based tagging.

4.  The appropriateness of the authors' own augmentation of CVE entries is a key factor in determining the value of this study.

4.1. Who are the 4 experts? The authors should add a description of the reason these persons are appropriate to perform such a task.

4.2.  The authors should introduce the contents of the standardized 137 approach proposed by Mapping MITRE ATT&CK to CVEs for Impact methodology [19].

4.3. The authors should elaborate on the content of a set of previously established common general guidelines in the manuscript.

Author Response

1. Thank you for the suggestion, it was a great idea! We added the article CVE + MITRE ATT&CK® to Understand Vulnerability Impact [11] and also integrated another related article [12] regarding improvements to vulnerability management:

“The ATT&CK matrix supports a better understanding of vulnerabilities and what an attacker could achieve by exploiting a certain vulnerability. ATT&CK technique details, such as detection and mitigation, are useful for system administrators, SecOps, or DevSecOps teams to obtain an assessment risk report in a short period of time while generating a remediation plan for discovered vulnerabilities. The  Center for Threat-Informed Defense team has created a very useful methodology [11] that helps the community build a more powerful threat intelligence database. The organization's defender team has to understand how important it is to bridge vulnerability and threat management with the adoption of this methodology as more reliable and consistent risk assessment reports will be obtained [12].”

  1. Additional details are presented in the introduction considering vulnerability management and risk assessment:

“Baker [11] highlights the importance of combining CVEs with the ATT&CK framework to achieve threat intelligence. Years ago, it was considerably harder for security teams to understand the attack surface, thus reducing their capacity to protect the organization against cyber attacks. With the emergence of the ATT&CK project, the security teams have a better overview of the CVEs based on known attack techniques, tactics, and procedures.

Vulnerability management can be divided into three categories, namely: the "Find and fix" game, the "Vulnerability risk" game, and the "Threat vector" game. The first one is a traditional approach where the vulnerabilities are prioritized by CVSS Score; this is applicable for small organizations with less dynamic assets. The second category consists of risk-based vulnerability management where organizational context and threat intelligence (such as CVE exploited in the wild properties) are considered; this applies to organizations that have security teams, but the number of CVEs is too large. The "Threat Vector" game includes the understanding of how the hackers might exploit the vulnerabilities while accounting for the MITRE ATT&CK framework mappings between CVEs and techniques, tactics, and procedures. The third category is the most efficient model of threat intelligence, with inputs delivered to the vulnerability risk management process from cyber attacks that have occurred and are trending. As such, security teams should take into account risks for building the vulnerability management program, but also threat intelligence to have a better understanding of vulnerabilities and to discover the attack chains within the network [13].”

  1. We introduced additional details in the Introduction:

“Even though linking CVEs to MITRE ATT&CK Enterprise Matrix would add massive value to the cybersecurity community, these two powerful tools are currently separated. However, manually mapping all 189,171 [10] CVEs currently recorded to one or more of the 192 different techniques in the MITRE ATT&CK Enterprise Matrix is a non-trivial task and the need for automated models emerges to map all existing entries to corresponding techniques. In addition, even if new CVEs would be manually labeled, an initial pre-labeling using a machine learning model before expert validation would be time effective and beneficial. Moreover, the model would provide technique labeling for zero-day vulnerabilities, which would be extremely helpful for security teams.”

  1. In order to avoid major alterations of the content, we used simple recipes in augmentation of CVE process. Additional details are presented in the Data Augmentation sub-section:

“We chose the EasyDataAugmenter (EDA) for augmenting the CVE dataset, which performs four simple, but powerful operations on the input texts: synonym replacement, random insertion, random swap, and random deletion. EDA significantly boosts performance and shows particularly strong results for smaller datasets [29, which makes it the perfect candidate for oversampling our labeled CVE corpus. Moreover, EDA does not perform major alterations of the content and is not as computationally expensive as other recipes, such as CLAREAugmenter, while providing satisfactory results on our CVE corpus.”

4.1 We added a description of the 4 experts in the Data Collection subsection.

“The labeling was performed by three 4th year undergraduate students in Computer Science with background courses in security, networking, and operating systems, and 1 Ph.D. student in Computer Science with 5+ years of experience in information security in the industry who provided guidance and helped reach consensus. The entire annotation process was overseen by a professor in cyber security.“

4.2 We provided extra details about the methodology in the Data Collection subsection.

“The Mapping MITRE ATT&CK to CVEs for Impact methodology consists of three steps. The first one is to identify the type of vulnerability (e.g., cross-site scripting, buffer overflow, SQL injection) based on the vulnerability type mappings. The next step is to find the functionality to which the attacker gains access by exploiting the CVE. The final step refers to determining the exploitation technique using the provided tips that offer details about the necessary steps to exploit a vulnerability.”

4.3 We added details about the guidelines used in creating the dataset in the Data Collection subsection.

“Our methodology started from these steps and added other common general guidelines before labeling the tactics and techniques, like searching for more details about a CVE on security blogs to get more relevant insights, or analyzing databases (e.g., the Vulnerability Database [22] and the Exploit Database - Exploits for Penetration Testers, Researchers, and Ethical Hackers [23]) for useful inputs about CVEs.”

Reviewer 3 Report

Authors amied to build an automated model that links a CVE to one or more MITRE ATT&CK Enterprise Matrix techniques, based on the text description provided in the CVE metadata. The paper is well-written and be helpful for researchers. 

Paper can be proofread.

Author Response

(The authors gave the same response as above.)

Reviewer 4 Report

It suggest re-writing the abstract to highlight the contribution and novelty.

Author Response

Thank you kindly for your comment, we highly appreciate it! We have modified the abstract accordingly to better highlight our contributions.

“Since cyber-attacks are ever-increasing in number, intensity, and variety, a strong need for a global, standardized cyber-security knowledge database has emerged as a means to prevent and fight cybercrime. Attempts already exist in this regard. The Common Vulnerabilities and Exposures (CVE) list documents numerous reported software and hardware vulnerabilities, thus building a community-based dictionary of existing threats. MITRE ATT&CK Framework describes adversary behavior and offers mitigation strategies for each reported attack pattern. While extremely powerful on their own, the tremendous extra benefit gained when linking these tools cannot be overlooked. This paper introduces a dataset of 1,813 CVEs annotated with all corresponding MITRE ATT&CK techniques and proposes models to automatically link a CVE to one or more techniques based on the text description from the CVE metadata. We establish a strong baseline that considers classical machine learning models and state-of-the-art pre-trained BERT-based language models while counteracting the highly imbalanced training set with data augmentation strategies based on the TextAttack framework. We obtain promising results as the best model achieved an F1-score of 47.84\%. In addition, we perform a qualitative analysis that uses Lime explanations to point out limitations and potential inconsistencies in CVE descriptions. Our model plays a critical role in finding kill chain scenarios inside complex infrastructures and enables the prioritization of CVE patching by the threat level. We publicly release our code together with the dataset of annotated CVEs.”

Round 2

Reviewer 2 Report

My comments in the last review round were adequately addressed.